

# High-performance pipeline for MutMap and QTL-seq

Yu Sugihara[1,2], Lester Young[3], Hiroki Yaegashi[1], Satoshi Natsume[1], Daniel J. Shea[1], Hiroki Takagi[4], Helen Booker[3,5], Hideki Innan[6], Ryohei Terauchi[1,2] and Akira Abe[1]

[1] Department of Genomics and Breeding, Iwate Biotechnology Research Center, Kitakami, Japan
[2] Graduate School of Agriculture, Kyoto University, Kyoto, Japan
[3] Department of Plant Sciences, University of Saskatchewan, Saskatoon, Saskatchewan, Canada
[4] Faculty of Bioresources and Environmental Sciences, Ishikawa Prefectural University, Nonoichi, Japan
[5] Department of Plant Agriculture, University of Guelph, Guelph, Ontario, Canada
[6] Graduate University for Advanced Studies, Hayama, Japan

## ABSTRACT

**Summary**. Bulked segregant analysis implemented in MutMap and QTL-seq is a powerful and efficient method to identify loci contributing to important phenotypic traits. However, the previous pipelines were not user-friendly to install and run. Here, we describe new pipelines for MutMap and QTL-seq. These updated pipelines are approximately 5–8 times faster than the previous pipeline, are easier for novice users to use, and can be easily installed through bioconda with all dependencies.
**Availability**. The new pipelines of MutMap and QTL-seq are written in Python and can be installed via bioconda. The source code and manuals are available online (MutMap: https://github.com/YuSugihara/MutMap, QTL-seq: https://github.com/YuSugihara/QTL-seq).

## INTRODUCTION

Bulked segregant analysis (*Michelmore, Paran & Kesseli, 1991*; *Giovannoni et al., 1991*; *Li & Xu, 2021*), as implemented in MutMap (*Abe et al., 2012*) and QTL-seq (*Takagi et al., 2013*), is a powerful and efficient method to identify loci contributing to important phenotypic traits. MutMap requires whole-genome resequencing of a single individual from the original cultivar and the pooled sequences of $F_2$ progeny from a cross between the original cultivar and mutant. MutMap uses the sequence of the original cultivar to polarize the site frequencies of neighboring markers and identifies loci with an unexpected site frequency, simulating the genotype of $F_2$ progeny.

QTL-seq was adapted from MutMap to identify single or multiple loci contributing to important phenotypic traits. It utilizes sequences pooled from two segregating progeny populations with extreme opposite traits (*e.g.*, resistant vs. susceptible to a pathogen) and single whole-genome resequencing of either of the parental cultivars. The original QTL-seq algorithm assumes that loci controlling phenotypic traits fix in opposite directions in

Corresponding author
Akira Abe, a-abe@ibrc.or.jp

two bulked populations through self-fertilizing. Therefore, QTL-seq is usually applied to homozygous genomes of the self-fertilizing plant but not to heterozygous genomes obligated to outcross.

Despite their usefulness, these programs are not user-friendly to install or run and require multiple user inputs. Another problem is that the programs requires Coval (*Kosugi et al., 2013*) for variant calling, which relies on the older versions of SAMtools (before 0.1.8). Updated software including PyBSASeq (*Zhang & Panthee, 2020*) and QTL-seqr (*Mansfeld & Grumet, 2018*) have been developed (*Li & Xu, 2021*).

In this study, we describe newly developed pipelines for MutMap and QTL-seq with updated features.

## IMPLEMENTATION

The new pipelines support read trimming by Trimmomatic (*Bolger, Lohse & Usadel, 2014*), replacing fastx-toolkit in the previous pipeline. Trimmed reads are aligned by BWA-MEM (*Li & Durbin, 2009*), replacing BWA-SAMPE, BWA-ALN and Coval. Improperly paired reads and PCR duplicates are filtered by SAMtools (*Li et al., 2009*). Subsequently, a VCF file is generated by the ''mpileup'' command implemented in BCFtools (*Li, 2011*). The user can start the analysis from any point in the process, *e.g.*, from raw FASTQs, trimmed FASTQs, BAM files, or a VCF file. MutPlot and QTL-plot, which are standalone programs, were developed for postprocessing of VCF files. Low-quality variants in a VCF file are filtered out based on mapping quality and strand bias and the actual and expected SNP-indices calculated based on the AD (allele depth) value of each sample pool (*Abe et al., 2012*). In QTL-seq, a ΔSNP-index is calculated by subtracting the SNP-index of one bulk from the other (*Takagi et al., 2013*). As an option, multiple testing correction (*Huang et al., 2020*) was also adopted to the simulation. Both pipelines ignore the SNPs which are missing in the parental sample. Candidate causal mutations in the VCF file are shown graphically after optionally executing SnpEff (*Cingolani et al., 2012*), which assesses the impact of located mutations on putatively expressed genes. The procedures are connected by a Python script.

## METHODS

To compare the performance of the new and old pipelines, we ran MutMap and QTL-seq on an AMD EPYC 7501 processor (Base 2.0 GHz) with 48 GB RAM and 12 threads (located at ROIS National Institute of Genetics in Japan). MutMap was run for two datasets, dataset 1 and dataset 2, used in the previous research (*Abe et al., 2012*). The original rice cultivar of both datasets was Hitomebore. The mutant bulks for dataset 1 and dataset 2 were Hit1917-pl and Hit1917-sd, respectively. These datasets can be downloaded as DRR004451 (Hitomebore), DRR001785 (Hit1917-pl), and DRR001787 (Hit1917-sd). MutMap v2.3.2 was run with the option ''-n 20'' as both mutant bulks contain 20 lines. The other parameters of MutMap v2.3.2 were set as default. For both datasets, ''IRGSP-1.0'' was used as the reference genome.

QTL-seq was run for the two datasets, dataset 3 and dataset 4, used in the previous study (*Takagi et al., 2013*). Dataset 3 was obtained from recombinant inbred lines (RILs)

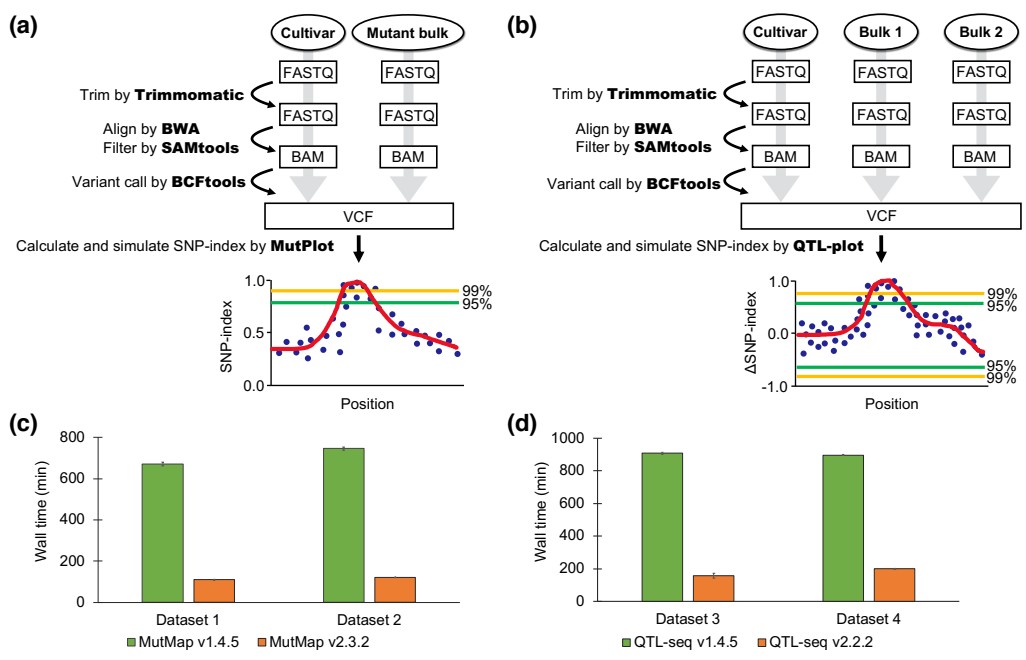

**Figure 1  Pipeline workflow and performance of MutMap and QTL-seq.** (A) The pipeline workflow of MutMap. (B) The pipeline workflow of QTL-seq. (C) Speed comparison between the new (v2.3.2) and old (v1.4.5) pipelines of MutMap. (D) Speed comparison between the new (v2.2.2) and old (v1.4.5) pipelines of QTL-seq. The values are mean ± SD ($n = 3$).

derived from a cross between Hitomebore and Nortai. Dataset 4 was obtained from $F_2$ progeny derived from a cross between Hitomebore and WRC57. We used a rice cultivar Hitomebore as a parental sequence for both datasets. These datasets can be downloaded as DRR004451 (Hitomebore), DRR003237 and DRR003238 (RILs derived from $F_1$ between Hitomebore and Nortai), and DRR003341 and DRR003342 ($F_2$ progeny derived from $F_1$ between Hitomebore and WRC57). For dataset 3, we ran QTL-seq v2.2.2 with the options "-n1 20 -n2 20 -F 6" because both bulks contain 20 $F_6$ RILs. For dataset 4, we ran QTL-seq v2.2.2 with the option "-n1 50 -n2 50 -F 2" as both bulks contain 50 $F_2$ progeny. The remaining parameters of QTL-seq v2.2.2 were set to their default values. For both datasets, "IRGSP-1.0" was used as the reference genome.

## RESULTS AND CONCLUSIONS

The new MutMap and QTL-seq pipelines are approximately 5–8 times faster than the previous pipelines. The significantly reduced processing time of the updated pipelines was accomplished by utilizing more applications with parallel processing (Trimmomatic, SAMtools, and BCFtools) for different steps including SNP calling and by omitting the previously implemented creation of a consensus FASTA file (Fig. 1). The ability of the updated pipeline to use a wider range of input file formats reduced the time required for file-management and data handling and makes the software easier to use. Further time-savings were accomplished with the new pipeline by removing user interactions that

were required in the previous version. Although the number of SNPs plotted was slightly different, the results of the old version and the new version were similar or had slightly better confidence index values (Fig. S1).

The simulation-based statistical test was adopted as the default because it allows addressing substantial heterogeneity in read depth among SNPs without any assumptions of statistical distributions of SNP-indices. We also implemented multiple testing correction following the parameters in the previous research (*Huang et al., 2020*). However, the method described in *Huang et al. (2020)* requires biological information such as number of chromosomes, genome size, and total centimorgan, which are not available in the majority of organisms, hence severely restricting the applicability. As stated by *Li & Xu (2021)*, the role of bulked segregant analysis is to map the target QTLs as a primary test, regardless of the statistical threshold criteria.

Currently, these new pipelines can be installed through bioconda with all dependencies. The new pipelines of MutMap and QTL-seq have improved performance and are more user-friendly to install and run, making them very useful for the purpose of genetics studies.

## ACKNOWLEDGEMENTS

Computations were performed on the NIG supercomputer at ROIS National Institute of Genetics and ComputeCanada infrastructure (http://www.computecanada.ca).

### Funding

This work was supported by JSPS KAKENHI Grant Number 20H02962 to Akira Abe. Support for Lester Young and Helen Booker was through Saskatchewan Agriculture Ministry's Agriculture Development Fund Grants 20160222 and 20170199 and Agriculture and AgriFoods Canada's Diverse Field Crops Cluster ASC-05. The funders had no role in study design, data collection and analysis, decision to publish, or preparation of the manuscript.

### Grant Disclosures

The following grant information was disclosed by the authors:
JSPS KAKENHI: 20H02962.
Saskatchewan Agriculture Ministry's Agriculture Development: 20160222, 20170199.

### Competing Interests

The authors declare there are no competing interests.

### Author Contributions

- Yu Sugihara conceived and designed the experiments, performed the experiments, analyzed the data, prepared figures and/or tables, authored or reviewed drafts of the paper, and approved the final draft.

- Lester Young conceived and designed the experiments, performed the experiments, analyzed the data, authored or reviewed drafts of the paper, and approved the final draft.
- Hiroki Yaegashi, Satoshi Natsume and Daniel J. Shea performed the experiments, analyzed the data, authored or reviewed drafts of the paper, and approved the final draft.
- Hiroki Takagi, Helen Booker, Hideki Innan and Ryohei Terauchi conceived and designed the experiments, authored or reviewed drafts of the paper, and approved the final draft.
- Akira Abe conceived and designed the experiments, performed the experiments, analyzed the data, prepared figures and/or tables, authored or reviewed drafts of the paper, and approved the final draft.

## Data Availability

The source code and manuals are available at GitHub (MutMap: https://github.com/YuSugihara/MutMap, QTL-seq: https://github.com/YuSugihara/QTL-seq).

The data is available at DDBJ: DRR004451, DRR001785, DRR001787, DRR003237, DRR003238, DRR003341, DRR003342.

## Supplemental Information

Supplemental information for this article can be found online at http://dx.doi.org/10.7717/peerj.13170#supplemental-information.

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
