# Peer review of "High-performance pipeline for MutMap and QTL-seq"

_PeerJ, doi:10.7717/peerj.13170_

## Round 0.1 · original submission · Minor Revisions

Dear Dr. Sugihara and colleagues:

Thanks for submitting your manuscript to PeerJ. I have now received five independent reviews of your work, and as you will see, all are very favorable. Well done! Nonetheless, some reviewers raised some relatively minor concerns about the research, and areas where the manuscript can be improved. I agree with the reviewers, and thus feel that their concerns should be adequately addressed before moving forward.

Please address these concerns and resubmit your manuscript along with a response to these concerns/issues.

I look forward to seeing your revision, and thanks again for submitting your work to PeerJ.

Good luck with your revision,

-joe

·

Basic reporting

This MS describes a new pipeline for MutMap and QTL-seq. The contents are concise and clear, and well written in an easy-to-understand manner.

Experimental design

The programs explained are available from GitHub. They are easy to obtain and install.

Validity of the findings

MutMap and other methods developed by the authors are widely used in other studies. It should be beneficial to provide researchers with this type of pipeline.

Additional comments

I have reviewed this MS in another journal. The MS was already revised and I have no further comments. It just deserves publication.

·

Basic reporting

no comment

Experimental design

no comment

Validity of the findings

no comment

Additional comments

no comment

Reviewer 3 ·

Basic reporting

The paper is brief, succinct and well written but I do have some suggestions for improvement:

1) Line 32 - Please cite the original papers by Giovannoni et al., 1991; Michelmore et al., 1991

2) Line 33 - I recommend changing the language here and in the abstract (ie line 23) to not limit the method to "agronomically important traits". BSA and QTLseq et al is commonly used in many genetic studies across many biological systems. Changing this language to say something like "to identify loci contributing to important phenotypic traits". This will help the paper be more interesting for a broader readership.

3) Line 40- "QTL-seq was adapted from MutMap to identify quantitative trait loci". - but it is also effective and commonly used for mapping single gene traits.

4) Line 43 -
a) Maybe better to write "algorithm assumed a mostly homozygous genome"
b) also, briefly describe the issues with heterozygous references in the context of QTLseq
c) it is unclear from the manuscript if the "modified QTL-seq" is implemented in the new version of the sofware.

5) Line 47-48 - Since the original programs came out and due to the limitations other software has been released to address those limitations suggest adding something like "As such multiple alternative pipelines have been suggested over the years including PyBSASeq, QTLseqr and others well reviewed here: https://doi.org/10.1111/tpj.15646";

Line 64-65 - "In QTL-seq, a ΔSNP-index is calculated by 65 subtracting one SNP-index from the other" might be better as "subtracting the SNP-index of one bulk from the other"

Line 65 - the multiple correction option: Explain better how this is integrated in to the the software and where it improves the results also on the github page this option is not recommended for first use and is not clear how the species has an impact on multiple correction, please expand the explanation in the paper.

Experimental design

I understand the minimal format for the software note, but there is very limited explanation of the methods used to validate the new software.
A brief methods section could improve the paper for testing the new implementation.

Line 73 is the place to include a description of the the datasets and their location for download - not the figure legend.

Also briefly describe the method for testing exactly what parameters were used and set.

Validity of the findings

Line 75 and 82 - The improvement in speed is essentially due to updates caused by using other more modern software, not due to improvements in the authors method. I would focus on the improved installation experience and user interface rather than speed as a main result. If there are other more novel functions of the software it is worth discussing those in more detail. For example the use of multiple correction and heterozygous genomes? although as a scan the github help it's not clear if that option is actually implemented or not.

Additional comments

Overall a quick piece that will give users of the new pipeline a paper to cite.
The biggest improvement over the previous version of the pipeline is the installation using conda - this will undoubtedly make the software more accessible to users.
I think the methods need to be better described and some other minor suggestions need to be improved and clarified as above.

Reviewer 4 ·

Basic reporting

The manuscript is concise and easy to understand.

Experimental design

The design of the entire study is solid.

Validity of the findings

The authors optimized the MutMap and QTL-seq pipelines they developed previously in the manuscript. Many components of the updated pipelines are able to utilize multiple cores of contemporary CPUs for parallel computing, so their performance increases dramatically. The updated pipelines are more flexible as well and can handle various input files, from raw sequencing data to files generated in the intermediate steps. Additionally, the new pipelines can be installed with a single command; dependence requirements are handled via bioconda so users do not need to fish around the different components of the pipelines. These new pipelines should be useful for the genetic community.

Reviewer 5 ·

Basic reporting

Good

Experimental design

Good

Validity of the findings

It is good to see the improved pipelines for MutMap and QTL-seq. The developed pipelines are user-friendly, and their speed is 5-8 times faster than the old-version pipelines. I suggest to accept it.

Minors:
L106: Brassica rapa should be italic.
L107-108: The first letter each word in title should be lower case except the first word.
Figure S1-E, and G: Y-axle should be ΔSNP-index, do not use Delta(SNP-index).

---

## Round 0.2 · accepted · Accept

Dear Dr. Sugihara and colleagues:

Thanks for revising your manuscript based on the concerns raised by the reviewers. I now believe that your manuscript is suitable for publication. Congratulations! I look forward to seeing this work in print, and I anticipate it being an important resource for groups studying improvements to pipelines for MutMap and QTL-seq analyses. Thanks again for choosing PeerJ to publish such important work.

Best,

-joe